# 'You can't start a car when there's no petrol left': a qualitative study of patient, family and clinician perspectives on implantable cardioverter defibrillator deactivation

Holly Standing [1], Richard G Thomson,[2] Darren Flynn,[3] Julian Hughes,[4] Kerry Joyce,[2] Trudie Lobban,[5] Stephen Lord,[6] Dan D Matlock,[7] Janet M McComb,[6] Paul Paes,[8] Chris Wilkinson [2], Catherine Exley[2]

► Prepublication history and additional online supplemental material for this paper are available online. To view these files, please visit the journal online (http://dx.doi.org/10.1136/bmjopen-2020-048024).

For numbered affiliations see end of article.

**Correspondence to**
Professor Catherine Exley;
catherine.exley@newcastle.ac.uk

## ABSTRACT

**Objective** To explore the attitudes towards implantable cardioverter defibrillator (ICD) deactivation and initiation of deactivation discussions among patients, relatives and clinicians.

**Design** A multiphase qualitative study consisting of in situ hospital ICD clinic observations, and semistructured interviews of clinicians, patients and relatives. Data were analysed using a constant comparative approach.

**Setting** One tertiary and two district general hospitals in England.

**Participants** We completed 38 observations of hospital consultations prior to ICD implantation, and 80 interviews with patients, family members and clinicians between 2013 and 2015. Patients were recruited from preimplantation to postdeactivation. Clinicians included cardiologists, cardiac physiologists, heart failure nurses and palliative care professionals.

**Results** Four key themes were identified from the data: the current status of deactivation discussions; patients' perceptions of deactivation; who should take responsibility for deactivation discussions and decisions; and timing of deactivation discussions. We found that although patients and doctors recognised the importance of advance care planning, including ICD deactivation at an early stage in the patient journey, this was often not reflected in practice. The most appropriate clinician to take the lead was thought to be dependent on the context, but could include any appropriately trained member of the healthcare team. It was suggested that deactivation should be raised preimplantation and regularly reviewed. Identification of trigger points postimplantation for deactivation discussions may help ensure that these are timely and inappropriate shocks are avoided.

**Conclusions** There is a need for early, ongoing and evolving discussion between ICD recipients and clinicians regarding the eventual need for ICD deactivation. The most appropriate clinician to instigate deactivation discussions is likely to vary between patients and models of care. Reminders at key trigger points, and routine discussion of deactivation at implantation and during advance care planning could prevent distressing experiences for both the patient and their family at the end of life.

### Strengths and limitations of this study

► Implantable cardioverter defibrillator implants are increasing and existing recipients are ageing, which makes optimal end-of-life planning an important and topical issue.
► This is the first study of its kind to include both clinic observations and semistructured interviews.
► There was a robust recruitment strategy, with a large number of participants enrolled.
► However, this study lacks the perspectives of primary care clinicians.
► A limited number of patients who were actively considering deactivation was recruited.

## INTRODUCTION

Implantable cardioverter defibrillators (ICD) improve survival in patients at high risk of sudden cardiac death through delivering treatments to terminate life-threatening arrhythmia.[1–3] However, as people age the risk of non-arrhythmic modes of death, such as frailty and accumulating comorbidity, progressive heart failure or other terminal diseases such as cancer may dominate, and the potential benefit from an ICD is reduced.[4–6] More than one-third of hospitalised patients with an ICD experience a ventricular tachyarrhythmia within the last hour of life,[7] and therefore leaving the device active is likely to lead to shocks at the end of life, which may be distressing, painful and futile.[8–12] Advance care planning (ACP) aims to ensure that appropriate measures are in place to aid patients' transition to end-of-life care and 'a good death'.[13] Although guidelines suggest that ongoing ICD activation should be considered as part of a patient's care goals,[14 15] clinicians may be failing to engage patients sufficiently in ACP.[9 16]

A proactive approach recommended by international guidelines mandates deactivation discussion prior to implantation as a part of shared decision making.[14 17 18] This has the declared support of both clinicians and patients.[10 12 19] Yet there is evidence that this discussion does not routinely occur.[20 21] Failure to engage patients may put undue pressure on family members to make deactivation decisions, who make more than half of deactivation requests.[22] Previous work has identified that patients recall the positive, 'life-saving' attributes of ICD discussions more than pre-implant end-of-life ACP discussions.[23 24] In this study, we add to the existing evidence base by including non-participant observations in addition to semistructured interviews. The study provides a comprehensive exploration of patient, family member and clinician attitudes towards deactivation discussions. We also explored when and how best to approach and undertake such discussions.

## METHODS
### Study design
The study comprised two phases: clinical observations, and interviews with patients, family members and clinicians. All participants provided written informed consent, and the investigation conforms with the Declaration of Helsinki.[25] Additional detail of the methods are provided in the full report.[26]

### Patient and public involvement
Public and patient engagement was initiated prior to the development of the outline application, and their feedback guided the research question and study approach. The Founder and Trustee of Arrhythmia Alliance (TL) was a coapplicant and member of our study advisory group, and provided service user input into all stages, including coauthoring this manuscript and the report to the study funders. Two patient/carer representatives (with an interest in end-of-life issues) joined the initial project advisory group. The Arrhythmia Alliance and the North of England Cardiovascular Network reviewed the research proposal, and their comments were incorporated.

### Sampling and recruitment
#### Phase 1: observations
Non-participant observations[27] were conducted in clinics across three hospitals: one tertiary care (implanting) centre and two district general hospitals. Observations at the tertiary care centre were with two implanting cardiologists, and observations at district general hospitals were with two heart failure nurses and one non-implanting cardiologist. Observations allowed researchers (HS/KJ) to familiarise themselves with the clinical environment and decision-making about ICDs. Field notes informed the sampling strategy and interview schedules for phase 2.

Cardiologists identified patients attending to discuss ICD implantation. Opt-out consent was used for this phase; study information was mailed with appointment letters, explaining that a researcher might be present to observe their consultation. An opt-out slip was included, and patients were asked to present this on arrival if they chose not to participate.

### Phase 2: interviews
#### Clinician interviews
Clinicians were recruited from five hospitals in the north of England with interviews, conducted over the telephone or at the clinician's place of work, exploring: the referral pathway for ICD implantation; approaches to risk communication; current approaches to, and people involved in, decision making; and timing of deactivation discussions. Purposive sampling[28] was used to capture a range of perspectives from specialities involved in the care of ICD patients.

#### Patient and family member interviews
We sought to include a range of patient experiences from those considering ICD implantation to those recently deactivated, as well as bereaved family members of ICD recipients 4–18 months post bereavement. The interview schedule covered issues identified through the observations and a literature review (see online supplemental material), including understandings of, and feelings about, the ICD; decision making about implantation; whether, how and by whom deactivation had been discussed; and preferences for information and decision support.

Next-of-kin contact details for deceased patients with an ICD were identified and an information pack was mailed, including an introductory letter from the implanting cardiologist. The letter invited interested individuals to return a consent-to-contact form to the research team. When received, the researcher arranged an interview. Invitations offered the option of being interviewed with a friend or family member, or passing the invitation to someone more closely involved in the patient's care. Interviews were conducted at the participants' home or over the telephone and were audiorecorded, transcribed verbatim and anonymised.

### Data analysis
Data collection was guided by the constant comparative method.[29] Data collection and analysis ran concurrently. Analysis of early interviews informed the interview schedule for subsequent interviews, and analysis was revised as data collection progressed. We critically examined the different perspectives and experiences of those involved in decision making about deactivation of ICDs. NVivo V.8 (QSR International, Warrington, UK) was used for data management. Data were analysed by three experienced qualitative researchers (HS and KJ, with support from CE). Regular meetings were held with the rest of the research team and the advisory group to discuss themes derived from the data. All names are pseudonyms.

### Findings
Between July 2013 and January 2015, 38 observations of consultations led by heart failure nurses, non-implanting

**Table 1** Participant characteristics (interviews)

| Patients | n |
|---|---|
| Preimplantation | 13 |
| Declined ICD | 8 |
| Postimplantation | 21 |
| Deactivated | 2 |
| Total | 44 |
| **Bereaved relatives** | |
| Bereaved spouse | 4 |
| Bereaved spouse and daughter | 2 |
| Bereaved son and daughter-in-law | 1 |
| Total | **7** |
| Clinicians | |
| Implanting cardiologist | 5 |
| Non-implanting cardiologist | 5 |
| Arrhythmia nurse | 1 |
| Heart failure nurses (hospital and community) | 6 |
| Cardiac physiologists* | 4 |
| Health psychologists | 2 |
| Palliative care specialists | 6 |
| Total | 29 |
| Overall total | 80 |

*A cardiac physiologist is a clinical scientist, known as a device (ICD or pacemaker) technician in the USA.
ICD, implantable cardioverter defibrillator.

and implanting cardiologists were conducted. Consultations lasted between 10 and 30 min. Forty-four patient interviews were conducted in 33 men and 11 women, aged between 47 and 85 years. Most patients (n=34) had been offered an ICD for primary prevention (patients without a history of sustained ventricular tachycardia or sudden cardiac arrest),[18] the remainder (n=10) were for secondary prevention (with a history of sustained tachycardia or sudden cardiac arrest).[18]

Table 1 provides a breakdown of patient, bereaved relative (n=7) and clinicians (n=29). Patient and bereaved relative interviews ranged from 11 to 113 (mean 44) min, clinician interviews from 20 to 83 (mean 42) min.

Four key themes were identified: the current status of deactivation discussions; patients' perceptions of deactivation; responsibility for deactivation discussions and decisions; and timing of deactivation discussions.

## Current status of deactivation discussions

Clinicians recognised the importance of timely ICD deactivation to avoid patients experiencing unnecessary shocks towards the end of life; however, current management of ICD deactivation was considered to be suboptimal at times. Patients were not consistently engaged in deactivation discussions and when this did occur, the patient was often close to death. Failure to deactivate the ICD in a timely manner could result in stressful and upsetting situations.

> I think the most recent one that we've had on the ward […] the decision was left too late, the patient was getting shocked too frequently, and nobody had had the chat to the family to say, "Well this is the defibrillator doing all of this, distressing them, and what we can do about it". (Dr J, non-implanting cardiologist) the patient in my view at that stage was actually dead, or was on the verge of being dead and it [ICD] was still delivering… Then once it had been deactivated, the patient passed away in a matter of a few moments. It wasn't pleasant. (Nurse C, heart failure nurse)

Bereaved relatives felt that deactivation was often poorly handled. One suggested reason for this was clinicians' apparent reticence to deactivate the ICD prior to the patient being in the last hours of life.

> she [cardiologist] said his heart looked alright and she didn't feel it was time to deactivate it … She [cardiologist] said to me "Do you think that he's ready to be deactivated?" And at the time I suppose I didn't, I said no …So that wasn't a very nice experience. (Shirley Bereaved wife)

Despite having engaged in ACP with a heart failure nurse, the patient's preference for deactivation was questioned by the implanting cardiologist who deferred the decision to the patient's wife. Although the ICD was eventually deactivated, it was reported that this occurred only hours before death, following intervention by the heart failure nurse. The patient's daughter, who also participated in the interview, felt it had been inappropriate for the cardiologist to put her mother in a position where she was expected to make an immediate deactivation decision.

Some patients reported facing apparent resistance from clinicians when trying to express their preference for deactivation.

> What he [the cardiologist] actually said [was], 'well, you'll basically drop down dead as soon as I switch it off'. It was so unkind. I wouldn't dare say that to anyone. (Fred, ICD deactivated)

This reported response from the cardiologist caused the patient and his wife a great deal of distress, and misrepresented the function of the ICD. The cardiologist's actions were perceived as an attempt to mislead the patient regarding the outcome of deactivation, so he would reconsider his decision to decline further medical intervention. This patient had declined his cardiologist's recommendation to be placed on the waiting list for heart transplant owing to previous experiences with organ transplantation and the associated side effects; the patient accepted he was approaching end of life and wanted limited intervention. Rather than exploring the patient's reasons for requesting deactivation, this account would seem to demonstrate a failure to undertake Shared Decision Making (SDM).

## Patients' perceptions of deactivation

Understanding of, and feelings about, deactivation among patients and family members were variable. Some appeared very pragmatic:

> You can't fire, start a car when there's no petrol in it, yeah? … it's no good firing your starter motor, starting your car, when there's no petrol in it. (Janet, bereaved relative)

For these individuals, deactivation was not necessarily viewed negatively, it was a means of avoiding unpleasant and unnecessary shocks:

> There are the end-of-life considerations […] I'm aware that, that this could happen [deactivation]. I think, yes, if I was seriously ill and near the end-of-life. I wouldn't want the thing shocking me. (Isobel, post-implantation)

The knowledge that deactivation may be a possibility could even offer a sense of control. However, for some patients the idea of deactivation was of concern. The device could become a source of comfort, which patients wished to keep activated as long as possible. This was particularly apparent among those who had recently had their ICD implanted.

> Burt: If you were to take it off us now I'd miss it
>
> Interviewer: So [you] would never think of switching it off?
>
> Burt: No, no, no (Burt, post-implantation)

## Appropriate timing of deactivation discussions
### Preimplantation

Preimplantation offers the first opportunity to introduce deactivation. It was recognised that discussing deactivation could influence a patient's decision on implantation. A few patients indicated that, had they been aware of the ICD's potential to influence their mode of death and the possible need for future deactivation, this may have affected their decision making.

> If it had been mentioned, I might have well have thought about not having one (Adrian, post-implant)

Although most patients expressed a preference to be informed about deactivation prior to implantation, some clinicians were less convinced. They felt that discussions about deactivation and end-of-life, even when hypothetical, conflicted with perceived patient needs and expectations of doctor–patient encounters.

> I mean most patients don't want to talk about it [EOL]… they come to see you 'cause they want to get better. They don't want to be told, "Well, you're going to die." (Dr B, non-implanting cardiologist)

Introducing deactivation pre-implantation was viewed as an 'illogical juxtaposition', which might give mixed messages and cause distress. ICDs are offered as a potentially lifesaving intervention; to discuss end of life was seen as inappropriate and potentially confusing. Clinicians also expressed concerns that patients might develop misconceptions that deactivation would result in immediate death.

> I suppose also this is something to do with your heart and how people comprehend that, so it seems it can be a lot more immediate and a lot more life-threatening even at the end-of-life. (Dr D, palliative care clinician)

Questions were also raised about patients' ability to remember information about deactivation given preimplantation. Preimplantation is often a time of high stress, when patients are expected to digest a considerable amount of information. Expecting patients to absorb and process information about deactivation may be unrealistic.

> But I don't think that people at that point [implantation] can really take on board that at some distant point in the future that they might need it changing (Dr E, palliative care clinician)

### Ongoing and evolving discussions

Rather than one-off conversations, both patients and clinicians indicated that deactivation should be an ongoing and evolving discussion, where the appropriateness of the ICD and how it fits with the patient's life is regularly reassessed. It was thought that 'sowing the seed' of deactivation pre-implantation might facilitate later conversations.

> you have to sow the seed, you know, plant the seed about what might happen in the future. (Nurse C, heart failure nurse)

The point of referral for an ICD could be an initial prompt to introduce the concept of deactivation, which could be revisited at intervals. One participant suggested that 'key points in admissions' could be used as triggers for deactivation discussions. Hospital and hospice admissions, and ICD firing or replacement, could be prompts for deactivation discussions. Inclusion of the patient on palliative care registers and production of ACP documentation, such as a do not attempt cardiopulmonary resuscitation form, could be used as final trigger points where deactivation must be discussed. This approach could avoid patients 'falling through the gaps', ensuring that every patient is engaged in deactivation discussions in a timely and appropriate manner.

### Responsibility for deactivation discussions and decisions

Heart failure nurses, implanting cardiologists, physiologists, palliative care clinicians and primary care clinicians may all have a role in both advance deactivation discussions and the ultimate decision to deactivate the ICD. However, no single clinical group currently took primary responsibility for these tasks.

Although cardiologists possess a high level of expertise that may facilitate them engaging patients in these

discussions, their role is traditionally focused on saving patients.

> I think doctors don't like talking about dying, because it's a sign, it's a sign that you have failed (Dr F, implanting cardiologist)

Further, depending on the patient's condition, contact with the cardiologist following implantation of the device may be infrequent, and so their capacity to have an ongoing role in deactivation discussions may be limited. Cardiac physiologists have regular contact with ICD patients, offering an opportunity to engage patients in ongoing discussions. However, some of the physiologists interviewed appeared reticent to engage in this conversation.

> So although I think we are in a good position in some ways, I think maybe there are other professionals out there who are in a position to do it as well. (Mr G, cardiac physiologist)

This reticence may be related to a feeling that they lacked the requisite skills to engage patients in SDM about continuation of ICD therapy. Cardiac physiologists' confidence and comfort with regards to these responsibilities may be improved by additional training. It is such a hard thing to do [discussing deactivation and death]. Probably need some training on how to broach the subject and when to broach it, and what to say. (Ms I, cardiac physiologist)

Heart failure nurses felt they were already undertaking the work of engaging patients in deactivation discussions.

> I have quite a few interesting conversations about death when I'm talking about defibrillators … it's just a simple sentence, 'your device will be switched off at some point, you can't live forever' (Nurse H, heart failure nurse)

Clinicians recognised that there are various professionals who could have a role in engaging patients in deactivation discussions.

> I think for any given patient somebody's going to be taking the lead, and it's not always clear-cut by role who that should be, so it may either be the GP [primary care physician], it's conceivably a heart failure nurse, it's conceivably a MacMillan [palliative care] nurse or our palliative care service (Dr M, palliative care clinician)

## DISCUSSION

Our findings demonstrate that clinicians understand the importance of deactivation discussions with patients and their families, and ACP for people with ICDs. Both patients and clinicians recognised the need for improvement; discussions about deactivation are often occurring late or not at all, resulting in negative impacts on patients and their families. We outline approaches for ensuring that these important and sensitive conversations take place ahead of time, based on extensive observations and semistructured interviews.

Previous work has shown that that patients report limited knowledge about deactivation, which is rarely discussed preimplantation.[23 24 30 31] Yet, SDM should involve bidirectional knowledge transfer between the patient and their families, and clinicians, including elicitation of individual patient preferences and values for different outcomes.[32] Indeed, informed consent requires that patients understand the risks, benefits and consequences of interventions in the light of what is important to them in making such decisions.[33] Our findings indicate a mismatch between patients' desire to be informed about deactivation and clinicians' perception of patients' information preferences, particularly pre-implantation. Many of our patient interviewees indicated they would be comfortable discussing deactivation and believed they should be informed about this before implantation so they had appropriate expectations of the future. Knowing about the possibility of future deactivation could afford patients a sense of control. However, while some clinicians perceived that patients did not want early deactivation discussion, and did not routinely pursue it during consultations, our results suggest that many patients do. Furthermore, the absence of such discussion may result in patient decisional regret and contribute to late decisions about deactivation. Another potential explanation is that clinicians feel uncomfortable discussing death in a consultation that is about preventing death. Knowing that many patients wish to be engaged in this discussion may help clinicians explore this preimplantation with more confidence.

There is evidence that clinicians experience greater discomfort regarding ICD deactivation than withdrawal of other life sustaining therapies,[34] and that ICDs are not considered in the same context as other end-of-life decisions.[11] It is possible that ethical questions about ICD deactivation as a form of euthanasia may contribute to some of the unease around initiating the relevant discussions.[9 35] Clinicians' reluctance to engage in end-of-life discussions may also stem from fear and anxiety of accepting the limits of their ability to save patients,[36] but perversely this may contribute to denying patients the opportunity of 'a good death'.

Our findings suggest a need for ongoing and evolving deactivation discussions, where the issue is introduced preimplantation (as desired by the majority of patients) and built on through subsequent encounters. Regularly revisiting ACP in relation to cardiac devices will enable clinicians to meet patients' goals of care better, recognising that these are likely to change over time with advancing years and the accumulation of health deficits.[37] However, we recognise that the increased use of remote device monitoring (in particular during the pandemic) have made this increasingly challenging. Prompts at appropriate points may help clinicians and patients to engage in these discussions,[38] and there is evidence that

education, clinical tools and standardised electronic health record templates are associated with increased rates of ICD deactivation and clinicians' confidence in managing the device.[39] The possibility of empowering patients and families to initiate the discussion themselves should also be considered.

Our study demonstrates a lack of consensus among clinicians regarding responsibility for deactivation discussions. Heart failure nurses often have established relationships and regular contact with patients and their families, which may facilitate them in aligning discussions with the patient's values and preferences, and building on the discussion over time. As others,[9 40] we suggest that a one-size-fits-all approach is not necessarily appropriate; the clinician most appropriate to discuss deactivation may differ between patients and at different times. Different models of care delivery for ICD patients will also have an influence; where specialist arrhythmia nurses are available, they will have an important role in deactivation discussions. However, these services were unavailable to the majority of our participants, resulting in more demand on the cardiologists and cardiac physiologists. Our findings support other research, which has suggested an interprofessional approach to initiating deactivation discussions.[41] Whoever is involved in care must attempt to ensure that patients do not fall through the gaps if clinicians fail to engage patients proactively in these discussions; and whichever clinicians are involved, they should have sufficient training, skills and confidence to undertake the task. Development of clear guidelines outlining the responsibility of each clinician group would provide clarity about expectations to engage in this work, and should support timely discussions within the multidisciplinary team and between patients, clinicians and their families. Additionally, we recognise that specialist-led secondary care clinics may lack access to comprehensive clinical information required for a holistic ongoing assessment of an individual and their circumstances. As such, we recognise the important role that general practitioners have in ACP, and the need for further data on how to optimise the interface between primary and secondary care with regard to ICD care.

## Strengths and limitations

This article is from a large qualitative study combining observations and interviews with a range of stakeholders. The observations allowed familiarity with the clinical setting; this also allowed us to refine our sampling strategies and interview schedule for the second phase of the study (interviews). The range of participants sampled provides insight into the perspectives of various clinician groups, as well as those of patients and their family members. However, we recognise the limitations of our work. Few interviews were conducted with patients with a deactivated ICD, or considering deactivation, because there is often a short time between deactivation and death.[31] Significant attempts were made to observe clinic appointments where decisions about ICD deactivation were made, and engage these patients in interviews, but success in recruiting this group was limited. Recruitment of bereaved relatives also proved challenging. The inclusion period for this group was expanded from 4 to 6 months following bereavement to 4–18 months, which improved our response rate. However, some of these participants may have been less able to recall events surrounding deactivation of the ICD (or not) and death, given the length of time post-bereavement. Finally, this study did not include primary care settings; future research should explore the views and experiences of primary care clinicians regarding their role in discussions and decision making around ICD deactivation.

## CONCLUSIONS

Our study has identified significant unmet need, particularly in the area of deactivation discussion. We believe that further work is needed in order to identify the best approach in supporting patients and clinicians to improving communication and care in this area. This is likely to include further education, development of multidisciplinary protocols and guidelines, and reminders for both clinicians and patients to have such discussions at key points.

**Author affiliations**
[1]Faculty of Health and Life Sciences, Northumbria University, Newcastle upon Tyne, UK
[2]Population Health Sciences Institute, Newcastle University, Newcastle upon Tyne, UK
[3]Centre for Rehabilitation, Exercise and Sports Science, Teesside University, Middlesbrough, UK
[4]Department of Population and Health Sciences, University of Bristol, Bristol, UK
[5]Arrhythmia Alliance, Chipping Norton, UK
[6]Newcastle Upon Tyne Hospitals NHS Foundation Trust, Newcastle Upon Tyne, UK
[7]School of Medicine, University of Colorado, Denver, Colorado, USA
[8]Northumbria Healthcare NHS Foundation Trust, North Shields, UK

**Acknowledgements** We would like to thank all the patients, relatives, and clinicians who allowed us to observe their consultations and took part in the interviews and workshops. We are grateful to the members of the project advisory group for their valuable advice throughout the study.

**Contributors** Developing the study protocol: CE, JH, KJ, TL, SL, DDM, JMM and RGT; Patient and public involvement: KJ; Developing the study design: CE, JH, KJ, TL, SL, DDM, PP and RGT; Data collection: HS, DF and KJ; Data interpretation: HS, CE, DF, JH, TL, SL, DDM, PP, CW and RGT. All authors were involved in drafting the work or revising it critically for important intellectual content. All authors approved the final version.

**Funding** This project was funded by The National Institute for Health Research Health Services and Delivery Research programme, (Project: 11/2004/29). Further information can be found at https://www.journalslibrary.nihr.ac.uk/programmes/hsdr/11200429/#/

**Competing interests** None declared.

**Patient consent for publication** Not required.

**Ethics approval** Ethical approval was granted by NHS Research Ethics Committee North East-Sunderland (Reference: 13/NE/0105).

**Provenance and peer review** Not commissioned; externally peer reviewed.

**Data availability statement** Data are available on reasonable request. Further details are available at https://pubmed.ncbi.nlm.nih.gov/27786432/

**ORCID iDs**
Holly Standing http://orcid.org/0000-0002-7806-8596
Chris Wilkinson http://orcid.org/0000-0003-0748-0150

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
