## [Reviewer comments · BMJ Open]

ARTICLE DETAILS

TITLE (PROVISIONAL)	'You can't start a car when there's no petrol left': A qualitative study of patient, family, and clinician perspectives on implantable cardioverter defibrillator deactivation
AUTHORS	Standing, Holly; Thomson, Richard; Flynn, Darren; Hughes, Julian; Joyce, Kerry; Lobban, Trudie; Lord, Stephen; Matlock, Dan D.; McComb, Janet M; Paes, Paul; Wilkinson, Chris; Exley, Catherine

VERSION 1 – REVIEW

REVIEWER	Sears, Samuel F. East Carolina University, Psychology
REVIEW RETURNED	16-Feb-2021

GENERAL COMMENTS	The current paper addresses a common clinical problem for cardiovascular clinics regarding patient understanding and family engagement in ICD activation. This qualitative study with 38 observations from approximately 80 people related to ICD use highlights major themes identified in the literature previously. Specifically, the major themes of patient perceptions of the activation, who should be responsible, and timing of deactivation discussions were highlighted. The current manuscript does a nice job of recreating the real-world clinical environment that these discussions take place in. It has ecological validity in its presentation. Previous committee reviews had concluded that more conversations about end of life should occur including in the pre implant period. However, as I visit other clinics and my own clinical practice, I do not see that occurring. In fact, I find it exactly as described in the paper as illogical juxtaposition. It presents a situation in which the clear, clinical ideal is very difficult to accomplish. This qualitative data set pushes the clinician further toward understanding that patients want that discussion throughout the process including the pre implant. Maybe more discussion about a "good death" in which the ICD is part could be described. This is the higher order "need" being addressed by this paper. Qualitative studies have an important place in providing a full color view of the clinical encounter. This manuscript accomplishes that very nicely-- the quotations are actually accurate. I would suggest that they have a quotation from each of the clinicians involved. In my account, I do not see health psychologists represented and I would like to see an addition from their group. In addition, the psychological aspects of decision making and fear in ICD patients
--

	may need to be reviewed and included in the manuscript in either the introduction or the discussion. What extent did shock history affect ratings and quotes? Investigators may want to discuss comorbidities in ICD patients as they are often the culprit leading to device deactivation. If they did not find that in their qualitative review, they should state that. Treatment burdens and medications associated with CHF, NIDDM, COPD, Frailty are consequential here beyond the ICD. The teasing apart of treatment burden and treatment futility are the crossroads. May want to consider how remote monitoring makes this process harder. Pts have less contact with ICD team, particularly post COVID. Stronger conclusions/directives: I really like the implication that each treatment team may need to discuss who, what, where, when on this issue as discussed on pg 19. Most teams have that as a default but this paper may want to take this further to formalize it via a “team meeting.” What else needs to be done to study this issue? Overall, I think this is a strong paper. I feel it is a touch long and brevity should be sought where possible.
--	--

REVIEWER	Fothergill, Rachael London Ambulance Service NHS Trust
REVIEW RETURNED	26-Feb-2021

GENERAL COMMENTS	Thank you for this very well written and extremely interesting paper. I just have one point for clarification, which I believe would be beneficial for the authors to address: I can see that ethical approval was obtained, but am also interested to know if Confidentiality Advisory Group (CAG) approval was sought relating to the use of deceased patient records to identify next of kin (where consent, of course, cannot be obtained). A statement relating to whether this approval was obtained, or if not, why it wasn't necessary, would be helpful.
---

VERSION 1 – AUTHOR RESPONSE

Reviewer 1. Professor S F Sears, East Carolina University

The current paper addresses a common clinical problem for cardiovascular clinics regarding patient understanding and family engagement in ICD activation. This qualitative study with 38 observations from approximately 80 people related to ICD use highlights major themes identified in the literature previously. Specifically, the major themes of patient perceptions of the activation, who should be responsible, and timing of deactivation discussions were highlighted.

The current manuscript does a nice job of recreating the real-world clinical environment that these discussions take place in. It has ecological validity in its presentation.

Previous committee reviews had concluded that more conversations about end of life should occur including in the pre implant period. However, as I visit other clinics and my own clinical practice, I do not see that occurring. In fact, I find it exactly as described in the paper as illogical juxtaposition. It presents a situation in which the clear, clinical ideal is very difficult to accomplish. This qualitative data set pushes the clinician further toward understanding that patients want that discussion throughout the process including the pre implant. Maybe more discussion about a “good death” in which the ICD is part could be described. This is the higher order “need” being addressed by this paper.

Many thanks for these comments. We have amended the introduction to recognise the importance of ‘a good death’, and added the following to the discussion:

Clinicians’ reluctance to engage in end-of-life discussions may also stem from fear and anxiety of accepting the limits of their ability to save patients, but perversely this may contribute to denying patients the opportunity of ‘a good death’.

Qualitative studies have an important place in providing a full color view of the clinical encounter. This manuscript accomplishes that very nicely-- the quotations are actually accurate. I would suggest that they have a quotation from each of the clinicians involved. In my account, I do not see health psychologists represented and I would like to see an addition from their group. In addition, the psychological aspects of decision making and fear in ICD patients may need to be reviewed and included in the manuscript in either the introduction or the discussion. What extent did shock history affect ratings and quotes?

We are grateful for this positive feedback. The quotations that we include in the manuscript are intended to be illustrative of the themes that we identified, rather than of each clinician, which was important in order to preserve the anonymity of contributors from small groups (which included health psychologists).

Investigators may want to discuss comorbidities in ICD patients as they are often the culprit leading to device deactivation. If they did not find that in their qualitative review, they should state that. Treatment burdens and medications associated with CHF, NIDDM, COPD, Frailty are consequential here beyond the ICD. The teasing apart of treatment burden and treatment futility are the crossroads. May want to consider how remote monitoring makes this process harder. Pts have less contact with ICD team, particularly post COVID.

Many thanks for these important points – we entirely agree, and have amended the discussion as follows:

‘Our findings suggest a need for ongoing and evolving deactivation discussions, where the issue is introduced pre-implantation (as desired by the majority of patients) and built upon through subsequent encounters. Regularly revisiting ACP in relation to cardiac devices will enable clinicians to meet patients’ goals of care better, recognising that these are likely to change over time with advancing years and the accumulation of health deficits.³⁷ However, we recognise that the increased use of remote device monitoring (in particular during the pandemic) have made this increasingly challenging.’

I really like the implication that each treatment team may need to discuss who, what, where, when on this issue as discussed on pg 19. Most teams have that as a default but this paper may want to take this further to formalize it via a “team meeting.”

The importance of a team approach has now been highlighted further in the discussion:

'Development of clear guidelines outlining the responsibility of each clinician group would provide clarity about expectations to engage in this work, and should support timely discussions within the multi-disciplinary team and between patients, clinicians, and their families.'

What else needs to be done to study this issue? Overall, I think this is a strong paper. I feel it is a touch long and brevity should be sought where possible.

We are grateful to you for your positive comments, and suggestions for improvement. We have edited the manuscript further in order to seek brevity, but without compromising content. We highlight studies including primary care clinicians as a key priority for future study – and have made our future recommendations more clear in the conclusion, which now reads:

'We believe that further work is needed in order to identify the best approach in supporting patients and clinicians to improving communication and care in this area. This is likely to include further education, development of multi-disciplinary protocols and guidelines, and reminders for both clinicians and patients to have such discussions at key points.'

Reviewer 2. Professor R Fothergill, London Ambulance Service NHS Trust

Thank you for this very well written and extremely interesting paper. I just have one point for clarification, which I believe would be beneficial for the authors to address:

I can see that ethical approval was obtained, but am also interested to know if Confidentiality Advisory Group (CAG) approval was sought relating to the use of decreased patient records to identify next of kin (where consent, of course, cannot be obtained). A statement relating to whether this approval was obtained, or if not, why it wasn't necessary, would be helpful.

We are very grateful for these comments. The research team were not processing confidential patient or social care information without patient consent, and therefore CAG approval was not required. The clinical team made the initial approach, and willing relatives completed a 'consent to contact' form.

Many thanks for considering our work for publication in BMJ Open.

VERSION 2 – REVIEW

REVIEWER	Fothergill, Rachael London Ambulance Service NHS Trust
REVIEW RETURNED	04-May-2021
GENERAL COMMENTS	Thank you for addressing my previous comment